

# Errors associated with calculating the gross nitrification rates in forested

# catchments using the triple oxygen isotopic composition ($\Delta^{17}$O) of stream nitrate

Weitian Ding[1], Urumu Tsunogai[1], Fumiko Nakagawa[1]

[1]Graduate School of Environmental Studies, Nagoya University, Furo-cho, Chikusa-

ku, Nagoya 464-8601, Japan

*Corresponding to*: Weitian Ding (ding.weitian.v2@s.mail.nagoya-u.ac.jp)



**Abstract**
A novel method for quantifying the gross nitrification rate (GNR) in each forested
catchment using the triple oxygen isotopic composition ($\Delta^{17}O$) of stream nitrate eluted
from the catchment has been proposed and applied in several recent studies. However,
the equations used in the calculations include the approximation that the $\Delta^{17}O$ value
of nitrate metabolized through either assimilation or denitrification within the forested
soil is equal to the $\Delta^{17}O$ value of nitrate in the stream. The GNR estimated from the
$\Delta^{17}O$ value of stream nitrate was more than six times the actual GNR in our simulated
calculation for a forested catchment where the nitrate in the soil exhibited $\Delta^{17}O$ values
larger than those in the stream while showing a decreasing trend with increasing
depths until that of stream nitrate at the bottom. As most of the reported soil nitrate in
forested catchments from past studies showed $\Delta^{17}O$ values higher than those of the
stream nitrate eluted from each catchment, we concluded that the GNR estimated
from the $\Delta^{17}O$ value of stream nitrate in the forested catchments was, to some extent,
an overestimate of the actual GNR.
**1 Introduction**
Nitrate ($NO_3^-$) is one of the important nitrogen nutrients for primary production in
forested ecosystems. Nitrification is the microbial process that produces $NO_3^-$ in each
forested ecosystem. Thus, quantifying the nitrification rate can assist in the evaluation
of the present and future states of each forest ecosystem. While the net nitrification
rate can be estimated from the increase in $NO_3^-$ concentration during a certain period,



the gross nitrification rate (GNR), which includes the net nitrification rate and the
metabolic rate of nitrate (e.g., assimilated by plants or decomposed through
denitrification), reflects the internal N cycling better than the net nitrification rate
(Bengtsson et al., 2003), especially in forested ecosystems, where the net nitrification
rate is negligible (Stark and Hart, 1997) while the metabolic rate is significant so that
the GNR often exceeds the net nitrification rate by order of magnitude (Verchot et al.,

2001).

Recently, several studies have successfully estimated GNR in water environments

such as lakes, using the $\Delta^{17}O$ values of $NO_3^-$ as a conserved tracer of the mixing ratio
between the atmospheric nitrate ($NO_3^-_{atm}$) deposited into the water environment and
the remineralized nitrate ($NO_3^-_{re}$) produced through nitrification therein (Tsunogai et
al., 2011, 2018). Although $NO_3^-_{re}$ always shows the $\Delta^{17}O$ values close to 0 ‰ because
its oxygen atoms derive from either terrestrial $O_2$ or $H_2O$ through nitrification,
$NO_3^-_{atm}$ always displays an anomalous enrichment in $^{17}O$ with $\Delta^{17}O$ values being
approximately $+26 \pm 3$ ‰ in Japan (Tsunogai et al., 2010, 2016) because of oxygen
transfers from atmospheric ozone (Michalski et al., 2003; Nelson et al., 2018).
Additionally, $\Delta^{17}O$ is almost stable during "mass-dependent" isotope fractionation
processes (Michalski et al., 2004; Tsunogai et al., 2016). Therefore, regardless of the
partial metabolism through denitrification or assimilation after deposition in a water
column, $\Delta^{17}O$ can be used as a conserved tracer of $NO_3^-_{atm}$ to calculate the mixing



ratio of $NO_3^-{}_{atm}$ to total $NO_3^-$ ($NO_3^-{}_{atm}/NO_3^-{}_{total}$) in a water column using the
following equation:
$[NO_3^-{}_{atm}]/[NO_3^-{}_{total}] = [NO_3^-{}_{atm}]/([NO_3^-{}_{re}] + [NO_3^-{}_{atm}]) = \Delta^{17}O/\Delta^{17}O_{atm}$      (1)
where $\Delta^{17}O_{atm}$ and $\Delta^{17}O$ denote the $\Delta^{17}O$ values of $NO_3^-{}_{atm}$ and $NO_3^-$ dissolved in
each water environment, respectively. Using both the $NO_3^-{}_{atm}/NO_3^-{}_{total}$ ratio estimated
from the $\Delta^{17}O$ value of $NO_3^-$ in a lake water column and the deposition rate of
$NO_3^-{}_{atm}$ into the lake, the GNR has been successfully estimated (Tsunogai et al., 2011,

2018).

In addition to application in water environments, the $\Delta^{17}O$ method has also been
applied to forested catchments for GNR determination (Fang et al., 2015; Hattori et
al., 2019; Huang et al., 2020). By using the deposition flux of $NO_3^-{}_{atm}$ into the
catchment as well as the elution flux of both unprocessed $NO_3^-{}_{atm}$ and $NO_3^-{}_{re}$ from the
stream, which can be determined from the $\Delta^{17}O$ values of $NO_3^-$ in stream water eluted
from the catchment, the GNR in each forested catchment has been estimated in a
manner similar to the estimation for the water environments (Fang et al., 2015).
Contrary to water environments, where the $\Delta^{17}O$ values of $NO_3^-$ within the water
layers are homogeneous and can be measured easily, it is often difficult to determine
the $\Delta^{17}O$ values of the $NO_3^-$ metabolized in soil layers. Consequently, past studies
approximated the values to be equal to the $\Delta^{17}O$ value of stream $NO_3^-$ eluted from
each forested catchment without actual observation (Fang et al., 2015, Hattori et al.,
2019, Huang et al., 2020). However, such an approximation should be conducted with



extreme caution, as the $\Delta^{17}O$ values of soil $NO_3^-$ are not always equal to those of the
stream (Hattori et al., 2019, Rose, 2014, Osaka et al., 2010). To clarify the details of
the approximation along with its impact on the final estimated GNR, we present an
accurate relationship between the $\Delta^{17}O$ of soil $NO_3^-$ and GNR, starting from the basic
isotope mass balance equations. Then, we present the GNR estimated for a forested
catchment in which the $\Delta^{17}O$ values of $NO_3^-$ in soil are measured. Finally, we
compare the GNR estimated in this study with the GNR estimated from the $\Delta^{17}O$
values of stream $NO_3^-$.

**2 Calculation**
The total mass balance equation of $NO_3^-$ including the GNR in each catchment can
be expressed as follows:
$$NO_3^-{}_{deposition} + GNR = NO_3^-{}_{leaching} + NO_3^-{}_{uptake} + GDR \qquad (2)$$
where $NO_3^-{}_{deposition}$, GNR, $NO_3^-{}_{leaching}$, $NO_3^-{}_{uptake}$, and GDR denote the deposition flux
of $NO_3^-$ into each catchment, gross nitrification rate in each catchment, leaching flux
of $NO_3^-$ from each catchment, uptake rate of $NO_3^-$ in each catchment, and gross
denitrification rate in each catchment, respectively.
The isotope mass balance for each $\Delta^{17}O$ value of $NO_3^-$ in the catchment can also be
calculated using the same method:
$$NO_3^-{}_{deposition} \times \Delta^{17}O(NO_3^-)_{atm} + GNR \times \Delta^{17}O(NO_3^-)_{nitrification} = NO_3^-{}_{leaching} \times \Delta^{17}O(NO$$
$$_3^-)_{stream} + NO_3^-{}_{uptake} \times \Delta^{17}O(NO_3^-)_{uptake} + GDR \times \Delta^{17}O(NO_3^-)_{denitrification} \qquad (3)$$



where $\Delta^{17}O(NO_3^-)_{atm}$, $\Delta^{17}O(NO_3^-)_{nitrification}$, $\Delta^{17}O(NO_3^-)_{stream}$, $\Delta^{17}O(NO_3^-)_{uptake}$, and
$\Delta^{17}O(NO_3^-)_{denitrification}$ denote the $\Delta^{17}O$ value of $NO_3^-{}_{atm}$ deposited into each
catchment, that of $NO_3^-{}_{re}$ produced through nitrification, that of $NO_3^-$ eluted from
each catchment, that of $NO_3^-$ assimilated by plants and other organisms in each
catchment, and that of $NO_3^-$ decomposed through denitrification in each catchment,
respectively.

If the $\Delta^{17}O$ values of $NO_3^-$ in the forested soil layers, where the $NO_3^-$ was

metabolized through either assimilation (by plants and other organisms) or
denitrification, are equal to the $\Delta^{17}O$ value of $NO_3^-$ in the stream, Eq. 4 can be
expressed as follows:
$\Delta^{17}O(NO_3^-)_{uptake} = \Delta^{17}O(NO_3^-)_{denitrification} = \Delta^{17}O(NO_3^-)_{stream}$                    (4)

Consequently, by combining Eqs. 3 and 4, we could obtain the following

relationship:
$NO_3^-{}_{deposition} \times \Delta^{17}O(NO_3^-)_{atm} + GNR \times \Delta^{17}O(NO_3^-)_{nitrification} = (NO_3^-{}_{leaching} + NO_3^-{}_{uptak}$
$_e + GDR) \times \Delta^{17}O(NO_3^-)_{stream}$                    (5)

We could estimate the GNR using Eq. 6 obtained from Eqs. 2 and 5 because we can

approximate the $\Delta^{17}O$ values of $NO_3^-{}_{re}$ produced through nitrification
($\Delta^{17}O(NO_3^-)_{nitrification}$) to be 0 (Michalski et al., 2003; Tsunogai et al., 2010):
$GNR = NO_3^-{}_{deposition} \times (\Delta^{17}O(NO_3^-)_{atm} - \Delta^{17}O(NO_3^-)_{stream})/\Delta^{17}O(NO_3^-)_{stream}$          (6)



Eq. 6 corresponds to the equation used in previous studies for quantifying the GNR
in each forested catchment (Eq. 4 in Fang et al., 2015; Eq. 8 in Hattori et al., 2019;
Eq. 4 Huang et al., 2020).

**3 Results and Discussion**
The $\Delta^{17}O$ values of $NO_3^-$ in forested soil layers should be equal to those of $NO_3^-$ in
the stream, as presented in Eq. 4 to obtain Eq. 6. While the number of simultaneous
observations of the oxygen isotopes of $NO_3^-$ in both the soil and stream in a given
forested catchment is limited (Hattori et al., 2019, Osaka et al., 2010, Rose, 2014), the
limited observations show that the oxygen isotopic ratios of soil $NO_3^-$ are mostly
higher than those of stream $NO_3^-$. For example, Hattori et al. (2019) reported that
more than 60 % of the soil exhibited $\Delta^{17}O$ values significantly higher than those of
stream $NO_3^-$ determined simultaneously ($\Delta^{17}O$ =+1 to +3 ‰). In addition, they found
a decreasing $\Delta^{17}O$ trend in soil $NO_3^-$ with depth, declining from greater than +20 ‰ at
the surface to less than +3 ‰ at depths of 25–90 cm from the surface. A similar
decreasing trend in the vertical distribution had been found in $\delta^{18}O$ in another forested
catchment, from greater than +35 ‰ at the surface soil to less than +10 ‰ at depths of
30–50 cm from the soil surface (Osaka et al., 2010). Besides, most of the soil $NO_3^-$
also exhibited $\delta^{18}O$ values higher than those in the stream (Osaka et al., 2010).
Furthermore, Rose (2014) monitored the horizontal distribution of $\Delta^{17}O$ of soil $NO_3^-$
by randomly setting 15 tension-free lysimeters at depths of 0–10 cm in a 39 ha



forested catchment, reporting $\Delta^{17}$O values significantly higher in soil $NO_3^-$ (+9.1 ±
5.8 ‰ on average) than in the stream $NO_3^-$ (+0.5 ‰ on average) eluted from the
forested catchment. As most of the root biomass is concentrated in the top 10 cm of
the soil in forested catchments (Jackson et al., 1996), most uptake reactions should
occur in that top 10 cm of soil. Consequently, the significant difference in the $\Delta^{17}$O
values between soil $NO_3^-$ and stream $NO_3^-$, particularly in the surface soil layers,
imply that the estimated GNRs in the forested catchment obtained from Eq. 6 were
inaccurate.
To demonstrate the impact of the differences in $\Delta^{17}$O between soil $NO_3^-$ and stream
$NO_3^-$ on the GNR, along with presenting the problems associated with the
approximation to obtain Eq. 6, we estimated the GNR for two simulated forested
soils—one with $NO_3^-$ showing a decreasing trend in $\Delta^{17}$O down to the $\Delta^{17}$O of stream
$NO_3^-$ (heterogeneous soil) (Fig. 1a and 1b) and one with $NO_3^-$ showing the same
$\Delta^{17}$O values as those of stream $NO_3^-$ (homogeneous soil) (Fig. 2a and 2b). With
Hattori et al. (2019) reporting the $NO_3^-{}_{deposition}$ as 7.0 kg of N ha$^{-1}$ y$^{-1}$, $NO_3^-{}_{leaching}$ as
2.6 kg of N ha$^{-1}$ y$^{-1}$, $\Delta^{17}$O($NO_3^-$)$_{atm}$ as +28.0 ‰, and $\Delta^{17}$O($NO_3^-$)$_{stream}$ as +2.2 ‰ in
their forested catchment study, we adopted the same values in our calculation.
We divided the soils in the heterogeneous forest soils into 10 layers in the vertical
direction, simulating the soils observed by Hattori et al. (2019), in which the $\Delta^{17}$O
values of $NO_3^-$ gradually decreased with increasing depth, varying from +28.0 to
+2.2 ‰ with a rate of decrease of +2.58 ‰ for each step (Fig. 1b). Similarly, we





assumed a gradual decrease with increasing depth in the leaching flux of $NO_3^-$, i.e.,
from 7 to 2.6 kg of N ha$^{-1}$ y$^{-1}$ with a rate of decrease of 0.44 kg of N ha$^{-1}$ y$^{-1}$ for each
step (Fig. 1c). In the homogeneous forest soils, we also divided the forested soils into
10 layers in the vertical direction. The change with depth in the leaching flux of $NO_3^-$
was the same as that in the heterogeneous soils (Fig. 2c), whereas the $\Delta^{17}O$ values of
$NO_3^-$ were constant at +2.2 ‰ in the soil layers (Fig. 2b).
Applying the total mass balance and isotope mass balance of $NO_3^-$ shown in Eqs. 2
and 3 to each layer, we estimated both the GNR (Figs. 1e and 2e) and total metabolic
rate of $NO_3^-$ (GDR + uptake) (Figs. 1d and 2d) in each layer assuming the following:
(1) $\Delta^{17}O$ values of $NO_3^-$ are constant in each layer, (2) vertical flow of $NO_3^-$ in the
soil layers proceed downward from the surface to the water layer with a uniform
residence time in each layer, and (3) the GNR and metabolic rate of $NO_3^-$ (GDR +
uptake) is zero in the water layer (layers beyond the no. 10 soil layer). Then, by
integrating the GNR determined for each layer, we estimated the total GNR in each
forested catchment.
Although the GNR estimated for the catchment with the homogeneous $\Delta^{17}O$ values
in soil $NO_3^-$ was 83.6 kg of N ha$^{-1}$ y$^{-1}$, exactly equal to that estimated by Hattori et al.
(2019) using Eq. 6 (Fig. 2e), the total GNR was a much smaller 13.0 kg of N ha$^{-1}$ y$^{-1}$,
simulated for the catchment with the heterogeneous $\Delta^{17}O$ values in soil $NO_3^-$ (Fig.
1e). Consequently, we conclude the following: (1) past studies estimating the GNR
using Eq. 6 approximated the $\Delta^{17}O$ value of soil $NO_3^-$ was homogeneous and always





equal to that of stream $NO_3^-$ mathematically and (2) the differences between the $\Delta^{17}O$
values of the soil $NO_3^-$ metabolized in a forested catchment and that of stream $NO_3^-$
resulted in a significant deviation in the GNR estimated using Eq. 6 from the actual
GNR.
Note that the linear variation in the leaching flux and $\Delta^{17}O$ values of soil $NO_3^-$ used
in the simulated calculations (Fig. 1) is just one of many possible variations in the
forested catchments. It is impossible to decide whether the linear variation was
realistic until the downward water flux, along with the concentration and $\Delta^{17}O$ values
of $NO_3^-$, is determined for each soil layer. However, the simultaneous observations of
the oxygen isotopes of soil $NO_3^-$ and stream $NO_3^-$ (Hattori et al., 2019; Osaka et al.,
2010; Nakagawa et al., 2018; Rose, 2014) imply that the approximation of the $\Delta^{17}O$
values of the soil $NO_3^-$ metabolized through assimilation or denitrification to be
always equal to the $\Delta^{17}O$ value of stream $NO_3^-$, shown in Fig. 2b, is unrealistic.
By combining the mass balance and isotope mass balance shown in Eqs. 2 and 3,
Eq. 7 can be obtained to accurately estimate the GNR:
$GNR = NO_3^-{}_{leaching} - NO_3^-{}_{deposition} + (NO_3^-{}_{deposition} \times \Delta^{17}O(NO_3^-)_{atm} -$
$NO_3^-{}_{leaching} \times \Delta^{17}O(NO_3^-)_{stream}) / \Delta^{17}O(NO_3^-)_{soil}$            (7)
where $\Delta^{17}O(NO_3^-)_{soil}$ denotes the $\Delta^{17}O$ values of $NO_3^-$ in forested soil, from which
the $NO_3^-$ was metabolized through either assimilation or denitrification. As most of
the soil $NO_3^-$ measured to date exhibit $\Delta^{17}O$ values higher than those of the stream
$NO_3^-$ eluted from each catchment (Hattori et al., 2019, Rose, 2014), the GNR





estimated from stream $NO_3^-$ using Eq. 6 is higher than the GNR estimated from soil

$NO_3^-$ using Eq. 7, to some extent. In other words, the GNR estimated from Eq. 6

overestimated the GNR in each forested catchment to some extent.

If we estimated the downward water flux at each soil layer, together with the $NO_3^-$

concentration and $\Delta^{17}O$ value of $NO_3^-$ in each soil layer using a tension-free lysimeter

(Inoue et al., 2021), we could estimate the vertical change in the leaching flux of

$NO_3^-$ for each soil layer along with the $\Delta^{17}O$ value of soil $NO_3^-$ in each layer. Then,

applying Eq. (7) in each layer, we can more accurately estimate the GNR for the

forested catchment by integrating the GNR estimated for each soil layer together with

a more accurate metabolic rate of $NO_3^-$ (GDR + uptake) of the forested catchment.

However, without such an observation of the distribution of the $\Delta^{17}O$ value of $NO_3^-$, it

is difficult to assume that the $\Delta^{17}O$ values of soil $NO_3^-$ are always equal to those of

stream $NO_3^-$; thus, the GNR should be reported with errors in which the possible

variations in the $\Delta^{17}O$ values of soil $NO_3^-$ are considered.

**4 Conclusion**

Past studies have proposed the $\Delta^{17}O$ method to determine the GNR in each forested

catchment. The equations used in the calculation presuppose that the $\Delta^{17}O$ values of

$NO_3^-$ in forested soils are homogeneous and equal to those of $NO_3^-$ in the stream;

however, in reality, the values are often heterogeneous and do not always equal to

those corresponding to the stream. It is essential to clarify/verify the $\Delta^{17}O$ values of



$NO_3^-$ in the forested soils and stream before applying the stream $NO_3^-$ $\Delta^{17}O$ values to
estimate the GNR.

*Data availability.* All data are presented in the Supplement.

*Author contributions.* WD, UT, and FN designed the study. WD and UT performed
data analysis and wrote the paper.

*Competing interests.* The authors declare that they have no conflict of interest.

*Acknowledgments*
We thank two anonymous referees, Dr. Joel Bostic, and Dr. Lucy Rose, for their
valuable remarks on an earlier version of this paper. We are grateful to the members
of the Biogeochemistry Group, Nagoya University, for their valuable support
throughout this study. This work was supported by a Grant-in-Aid for Scientific
Research from the Ministry of Education, Culture, Sports, Science, and Technology of
Japan under grant numbers 22H00561, 17H00780, 22K19846, the Grant-in-Aid for
JSPS Fellows under grant number 23KJ1088, the Yanmar Environmental
Sustainability Support Association, and the river fund of the river foundation, Japan.
Weitian Ding would like to take this opportunity to thank the "Nagoya University
Interdisciplinary Frontier Fellowship" supported by Nagoya University and JST, the



establishment of university fellowships towards the creation of science technology
innovation, Grant Number JPMJFS2120.

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



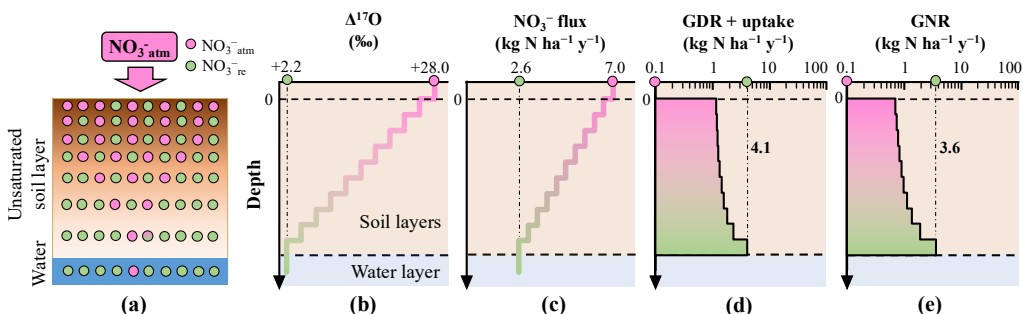

**Figure. 1**. Distribution of $NO_3^-{}_{atm}$ in the simulated forested soil where the distribution
of the $\Delta^{17}O$ values of $NO_3^-$ is heterogeneous (a). Vertical distribution of the following
parameters in the forested soil: the simulated $\Delta^{17}O$ values of $NO_3^-$ (b), simulated
leaching flux of $NO_3^-$ (c), estimated $NO_3^-$ consumption rate (GDR + uptake) (d), and
estimated GNR (e).

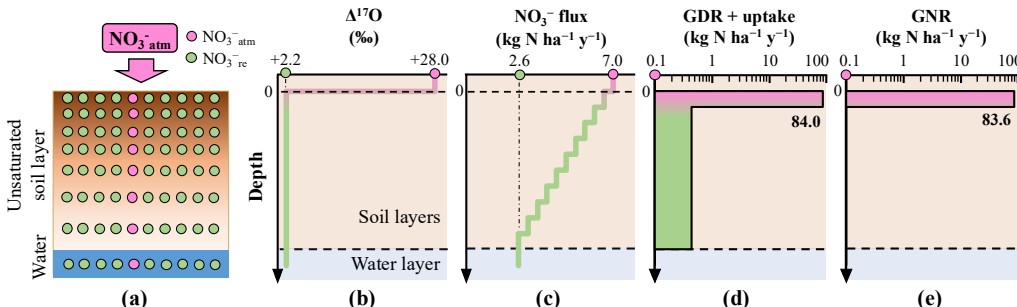

**Figure. 2**. Distribution of $NO_3^-{}_{atm}$ in the simulated forested soil where the distribution
of the $\Delta^{17}O$ values of $NO_3^-$ is homogeneous (a). Vertical distribution of the following
parameters in the forested soil: the simulated $\Delta^{17}O$ values of $NO_3^-$ (b), simulated
leaching flux of $NO_3^-$ (c), estimated $NO_3^-$ consumption rate (GDR + uptake) (d), and
estimated GNR (e).