# Peer review of "Bias in calculating gross nitrification rates in forested catchments using the"

_EGUsphere, 2023_

## Author Comment (AC1)

Dear Referee #1

Thank you very much for your valuable comments on our manuscript. We would like to respond to each of your comments one by one.

**Their simulation is based on a case where nitrate fluxes decrease with soil depth, which is the case if net nitrification is negative (nitrate consumption larger than gross nitrification). It would be very interesting to see what happens in the case of a positive net nitrification. This could be done with just one more simulation.**

Thank you for your comment. Our simulation was done for the forested catchment reported by Hattori et al. (2019). While the deposition flux of $NO_3^-$ was 7.0 kg of N $ha^{-1}$ $y^{-1}$, the leaching flux of $NO_3^-$ was 2.6 kg of N $ha^{-1}$ $y^{-1}$ in the forested catchment, so that the influx of $NO_3^-$ was higher than that of outflux. Thus, the $NO_3^-$ fluxes always decreased with soil depth in our original simulations, shown by Figures 1c and 2c in the manuscript.

In response to your request, we made a new simulated calculation, in which $NO_3^-$ fluxes increased with soil depth in the soil layers from 1 to 5 with an increasing rate of 0.44 kg of N $ha^{-1}$ $y^{-1}$ for each layer, while $NO_3^-$ fluxes decreased with soil depth in the soil layers from 6 to 10 with a decreasing rate of 1.32 kg of N $ha^{-1}$ $y^{-1}$ for each layer (Table R1). While the newly estimated GNR (19.1 kg of N $ha^{-1}$ $y^{-1}$) was comparable with that estimated for the forested catchment with the profile shown by Figure 1 (13.0 kg of N $ha^{-1}$ $y^{-1}$), it was still significantly smaller than the GNR calculated by using Eq.6 (83.6 kg of N $ha^{-1}$ $y^{-1}$). Such additional simulated calculation by changing the nitrate fluxes with soil depths further supports our conclusion that the GNR estimated from the $\Delta^{17}O$ value of stream nitrate in forested catchments was, to some extent, an overestimate of the actual GNR.

In addition, the present results imply that the most important parameter to determine total GNR (and thus total GDR + uptake) is the $\Delta^{17}O$ value of $NO_3^-$ consumed in soil layers. That is, the depth profile of $NO_3^-$ fluxes has little impact on GNR.

**Table R1.** $\Delta^{17}O$ values of $NO_3^-$, leaching flux of $NO_3^-$, total consumption rate of $NO_3^-$ (GDR + uptake), and GNR in the simulated forested soil where the distribution of $\Delta^{17}O$ values of $NO_3^-$ is heterogeneous. While the net nitrification from soil layer 1 to 5 showed positive values, the soil layer 6 to 10 showed negative values.

| Depth layer | $\Delta^{17}O$ ‰ | $NO_3^-$ flux | GDR +uptake | GNR |
|---|---|---|---|---|
| | | kg of N ha$^{-1}$ y$^{-1}$ | | |
| 0 | 28.0 | 7.0 | 0.0 | 0.0 |
| 1 | 25.4 | 7.4 | 0.3 | 0.7 |
| 2 | 22.8 | 7.9 | 0.4 | 0.8 |
| 3 | 20.2 | 8.3 | 0.6 | 1.0 |
| 4 | 17.7 | 8.8 | 0.8 | 1.2 |
| 5 | 15.1 | 9.2 | 1.1 | 1.5 |
| 6 | 12.5 | 7.9 | 3.2 | 1.9 |
| 7 | 9.9 | 6.6 | 3.4 | 2.1 |
| 8 | 7.3 | 5.2 | 3.6 | 2.3 |
| 9 | 4.7 | 3.9 | 4.2 | 2.9 |
| 10 | 2.2 | 2.6 | 6.0 | 4.7 |
| 11 | 2.2 | 2.6 | 0.0 | 0.0 |
| Total | | | 23.5 | 19.1 |

**In many soils, preferential water flow can be observed. In such cases, there is not a single, homogeneous nitrate pool per soil layer but nitrate that is more or less mobile along the flow paths and nitrate that is more bound within the soil matrix. The first is more prone to leaching and perhaps uptake, the second to denitrification. Simulating this would be a difficult task, probably out of the scope of the present article. Nevertheless, it would be useful if the authors would discuss this point**

Thank you for your comment. As you point out, the leaching flux of soil $NO_3^-$ in each layer is complex. Thus, we have added the following sentences in the manuscript (P10, L171-179).

Note that the linear variation in the leaching flux and $\Delta^{17}O$ values of soil $NO_3^-$ used in the simulated calculations is just one of many possible variations in the forested catchments. It is impossible to decide whether the linear variation was realistic until the downward water flux, along with the concentration and $\Delta^{17}O$ values of $NO_3^-$, is determined for each soil layer. However, the simultaneous observations of the oxygen isotopes of soil $NO_3^-$ and stream $NO_3^-$ (Hattori et al., 2019; Osaka et al., 2010; Nakagawa et al., 2018; Rose, 2014) imply that the approximation of the $\Delta^{17}O$ values of the soil $NO_3^-$ consumed through assimilation or denitrification to be always equal to the $\Delta^{17}O$ value of stream $NO_3^-$, shown in Fig. 2b, is unrealistic.

**and especially if they could make recommendations on how to sample nitrate from the soil for $\Delta^{17}O$ determination: zero-tension lysimetry, tension lysimetry, centrifugation, extraction? I'm not sure if clear answers can be given with the present knowledge of soil nitrate transformations, but at least the question would deserve to be raised.**

Thank you for your comment. We have recommended the sampling method of soil nitrate in the manuscript (P11, L191-194).

If we estimated the downward water flux at each soil layer, together with the $NO_3^-$ concentration and $\Delta^{17}O$ value of $NO_3^-$ in each soil layer using a tension-free lysimeter (Inoue et al., 2021), we could estimate the vertical change in the leaching flux of $NO_3^-$ for each soil layer along with the $\Delta^{17}O$ value of soil $NO_3^-$ in each layer.

**L. 3: the word "eluted" is rather used for the what is done on purpose in the lab. In this case, for the process observed in the nature, a better choice would probably be "leached".**

Thank you for your suggestion. We would like to change the "eluted" to "leached" in the revised manuscript.

**Line 6: instead "nitrate metabolized", it would be better to write "nitrate that is consumed", first because as soon as it is consumed, it is no longer nitrate, and second because "metabolized" is rather used to indicate that it is incorporated into organic matter, which is not the case for the denitrification.**

Thank you for your suggestion. We would like to change the "nitrate metabolized" to "nitrate that is consumed" in the revised manuscript.

**L. 24: on the same idea: "consumption" instead of "metabolic".**

Thank you for your suggestion. We would like to change the "metabolic" to "consumption" in the revised manuscript.

**L. 27: "is negligible" is too general, better add "often".**

Thank you for your suggestion. We would like to revise this in the revised manuscript.

**L. 28: "by order of magnitude": do you mean "one" order?**

We would like to revise the sentence to "the GNR often exceeds the net nitrification rate by several orders of magnitude."

**L. 21-29: very long sentence.**

We would like to revise the sentence in the revised manuscript.

**L. 31: it would be useful to explain shortly that the $\Delta$ anomaly is based on the $\delta$ of both $^{17}O$ and $^{18}O$ and that it is purposely defined so as to make it independent of mass-dependent fractionation.**

We would like to add the information in the revised manuscript.

**L. 31: in my opinion, "conservative" would be better than "conserved" (because it tends to be conserved but it is not always perfectly conserved).**

Thank you for your suggestion. We would like to change the "conserved" to "conservative" in the revised manuscript.

**L. 33: it seems strange to write "REmineralized" when it may be mineralized for the first time after centuries of N staying in the soil in the organic matter.**

We would like to revise this in the revised manuscript.

**L. 34-38, 53-57: long sentences.**

We would like to revise the sentences in the revised manuscript.

**L. 76: in this equation, some processes are denoted as subscript of NO$_3^-$ (like deposition) while others are denoted for themselves (like GNR). GNR and GDR are usually expressed as a nitrogen rather than as a nitrate flux. As it is written, the equation lets it open. It would be better to explicitly express all rates either as N or as NO$_3^-$.**

Thank you for your suggestion. Our simulation was done for the forested catchment reported by Hattori et al. (2019). Thus, the symbols used in the manuscript were in accordance with Hattori et al. (2019) as well. We would like to mention this in the revised manuscript.

**L. 74-80, 85-90: it is not clear why the word "each" is always used for the catchments (not only here, in general in the text).**

We would like to revise these in the revised manuscript.

**L. 112-113: repeated usage of the word "limited".**

We would like to revise this in the revised manuscript.

**L. 116: which one of the $\Delta^{17}O$ is this? Or is it the difference?**

The $\Delta^{17}O$ denotes the $\Delta^{17}O$ of stream nitrate. We would like to revise this in the revised manuscript.

**L. 127-128: fine roots would be much more relevant than the total root biomass (with coarse roots obviously overrepresented close to the stem and thus close to the surface).**

Thank you for your suggestion. We would like to add the information in the revised manuscript.

**L. 146-148: it may be useful to explain this as a gradual uptake (consumption) of nitrate as water moves down the profile.**

Thank you for your suggestion. We would like to add the information in the revised manuscript.

**L. 152-160: these assumptions are obviously simplifications compared to real measurements, but they make sense for the demonstration. It would be interesting to test also the assumption of nitrate fluxes increasing with depth because of a positive net nitrification.**

Thank you for your suggestion. We have simulated the positive net nitrification in the soil layers above.

**L. 175-179: long sentence.**

We would like to revise the sentence in the revised manuscript.

**L. 220: as written, it is like anonymous reviewers would be named, which does not make sense.**

We would like to revise the sentence in the revised manuscript.

**Fig. 1, fig. 2: the soil does not float above water and therefore "soil layers" and "water layer" should rather be marked "unsaturated soil layers" and either "water-saturated soil layer" or "seepage water" (as these two are considered to exhibit the same flux).**

We would like to revise the "soil layers" and "water layer" to "unsaturated soil layers" and "water-saturated soil layer" in the revised manuscript.

We would like to thank you for the helpful comments. We hope that our responses to your comments are satisfactory.

Sincerely,
Weitian Ding
PhD student
Graduate School of Environmental Studies,
Nagoya University
Furo-cho, Chikusa-ku, Nagoya,
464-8601, JAPAN
Phone: +81-70-4436-3157
E-mail: ding.weitian.v2@s.mail.nagoya-u.ac.jp
Cc: Drs. Urumu Tsunogai and Fumiko Nakagawa

**Reference**

Hattori, S., Nuñez Palma, Y., Itoh, Y., Kawasaki, M., Fujihara, Y., Takase, K., and Yoshida, N.: Isotopic evidence for seasonality of microbial internal nitrogen cycles in a temperate forested catchment with heavy snowfall, Science of the Total Environment, 690, 290–299, https://doi.org/10.1016/j.scitotenv.2019.06.507, 2019.

Inoue, T., Nakagawa, F., Shibata, H., and Tsunogai, U.: Vertical Changes in the Flux of Atmospheric Nitrate From a Forest Canopy to the Surface Soil Based on $\Delta^{17}O$ Values, Journal of Geophysical Research: Biogeosciences, 126, 1–18, https://doi.org/10.1029/2020JG005876, 2021.

Nakagawa, F., Tsunogai, U., Obata, Y., Ando, K., Yamashita, N., Saito, T., Uchiyama, S., Morohashi, M. and Sase, H.: Export flux of unprocessed atmospheric nitrate from temperate forested catchments: A possible new index for nitrogen saturation, Biogeosciences, 15(22), 7025–7042, doi:10.5194/bg-15-7025-2018, 2018.

Osaka, K., Ohte, N., Koba, K., Yoshimizu, C., Katsuyama, M., Tani, M., Tayasu, I., and Nagata, T.: Hydrological influences on spatiotemporal variations of $\delta^{15}N$ and $\delta^{18}O$ of nitrate in a forested headwater catchment in central Japan: Denitrification plays a critical role in groundwater , Journal of Geophysical Research: Biogeosciences, 115, n/a-n/a, https://doi.org/10.1029/2009jg000977, 2010.

Rose, L. A.: Assessing the nitrogen saturation status of appalachian forests using stable isotopes of nitrate [PhD thesis, University of Pittsburgh]. Retrieved from http://d-scholarship.pitt.edu/22783/1/LRose_ETD_081914_revised1.pdf.

---

## Author Response (AR1)

May 6, 2024

Dr. Frank Hagedorn
Editor of Biogeosciences

Title: Errors associated with calculating the gross nitrification rates in forested catchments using the triple oxygen isotopic composition ($\Delta^{17}$O) of stream nitrate
Authors: Weitian Ding et al.
MS No.: egusphere-2023-2753

Dear Dr. Frank Hagedorn:

Thank you very much for handling our manuscript. We would like to thank the referees as well for the constructive comments on our manuscript. We have carefully studied the comments and revised the manuscript accordingly. We include below point-by-point responses to the comments, and detailed descriptions of the modifications we made to the manuscript. Besides, we also uploaded the revised manuscript in MS Word, in which all the revisions from BGD version were recorded. We hope that with these changes you will find our revised manuscript appropriate for publication in your journal.

Sincerely yours,
Weitian Ding
Postdoctoral researcher
Graduate School of Environmental Studies,
Nagoya University
Furo-cho, Chikusa-ku, Nagoya,
464-8601, JAPAN
Phone: +81-70-4436-3157
E-mail: ding.weitian.v2@s.mail.nagoya-u.ac.jp
Cc: Drs. Urumu Tsunogai and Fumiko Nakagawa

**Response to the handing associate editor:**

**Here, I ask the authors to add 1-2 sentences to the discussion that in the case of the water environments, the Δ17O values and NO3 are mostly homogeneous in the water column due to the active vertical mixing, which is not the case in soils.**

Thank you for the advising. We added the following sentences in the revised manuscript (P4/L61-62; P7/L114-116).

Contrary to water environments, where the $\Delta^{17}O$ values of $NO_3^-$ in the water layers are homogeneous in the water column due to the active vertical mixing of water and can be measured easily, it is often difficult to determine the $\Delta^{17}O$ values of $NO_3^-$ consumed in soil layers.
Differ from water environments, vertical mixing of water/soil is difficult in forested soil, so the $\Delta^{17}O$ values of soil $NO_3^-$ are often heterogeneous.

**The manuscript requires an improved wording**

Thank you for the advising. We improved wording in the revised manuscript. Besides, the revised manuscript reviewed by an experienced editor whose first language is English and who specializes in editing papers written by scientists whose native language is not English.

**Response to the referee #1:**

**Their simulation is based on a case where nitrate fluxes decrease with soil depth, which is the case if net nitrification is negative (nitrate consumption larger than gross nitrification). It would be very interesting to see what happens in the case of a positive net nitrification. This could be done with just one more simulation.**

Thank you for your comment. Our simulation was done for the forested catchment reported by Hattori et al. (2019). While the deposition flux of $NO_3^-$ was 7.0 kg of N $ha^{-1}$ $y^{-1}$, the leaching flux of $NO_3^-$ was 2.6 kg of N $ha^{-1}$ $y^{-1}$ in the forested catchment, so that the influx of $NO_3^-$ was higher than that of outflux. Thus, the $NO_3^-$ fluxes always decreased with soil depth in our original simulations, shown by Figures 1c and 2c in the manuscript.

In response to your request, we made a new simulated calculation, in which $NO_3^-$ fluxes increased with soil depth in the soil layers from 1 to 5 with an increasing rate of 0.44 kg of N $ha^{-1}$ $y^{-1}$ for each layer, while $NO_3^-$ fluxes decreased with soil depth in the soil layers from 6 to 10 with a decreasing rate of 1.32 kg of N $ha^{-1}$ $y^{-1}$ for each layer (Table R1). While the newly estimated GNR (19.1 kg of N $ha^{-1}$ $y^{-1}$) was comparable with that estimated for the forested catchment with the profile shown by Figure 1 (13.0 kg of N $ha^{-1}$ $y^{-1}$), it was still significantly smaller than the GNR calculated by using Eq.6 (83.6 kg of N $ha^{-1}$ $y^{-1}$). Such additional simulated calculation by changing the nitrate fluxes with soil depths further supports our conclusion that the GNR estimated from the $\Delta^{17}O$ value of stream nitrate in forested catchments was, to an extent, an overestimate of the actual GNR.

In addition, the present results imply that the most important parameter to determine total GNR (and thus total GDR + uptake) is the $\Delta^{17}O$ value of $NO_3^-$ consumed in soil layers. That is, the depth profile of $NO_3^-$ fluxes has little impact on GNR.

**Table R1.** $\Delta^{17}O$ values of $NO_3^-$, leaching flux of $NO_3^-$, total consumption rate of $NO_3^-$ (GDR + uptake), and GNR in the simulated forested soil where the distribution of $\Delta^{17}O$ values of $NO_3^-$ is heterogeneous. While the net nitrification from soil layer 1 to 5 showed positive values, the soil layer 6 to 10 showed negative values.

| Depth layer | $\Delta^{17}O$ ‰ | $NO_3^-$ flux kg of N ha$^{-1}$ y$^{-1}$ | GDR +uptake | GNR |
|---|---|---|---|---|
| 0 | 28.0 | 7.0 | 0.0 | 0.0 |
| 1 | 25.4 | 7.4 | 0.3 | 0.7 |
| 2 | 22.8 | 7.9 | 0.4 | 0.8 |
| 3 | 20.2 | 8.3 | 0.6 | 1.0 |
| 4 | 17.7 | 8.8 | 0.8 | 1.2 |
| 5 | 15.1 | 9.2 | 1.1 | 1.5 |
| 6 | 12.5 | 7.9 | 3.2 | 1.9 |
| 7 | 9.9 | 6.6 | 3.4 | 2.1 |
| 8 | 7.3 | 5.2 | 3.6 | 2.3 |
| 9 | 4.7 | 3.9 | 4.2 | 2.9 |
| 10 | 2.2 | 2.6 | 6.0 | 4.7 |
| 11 | 2.2 | 2.6 | 0.0 | 0.0 |
| Total | | | 23.5 | 19.1 |

**In many soils, preferential water flow can be observed. In such cases, there is not a single, homogeneous nitrate pool per soil layer but nitrate that is more or less mobile along the flow paths and nitrate that is more bound within the soil matrix. The first is more prone to leaching and perhaps uptake, the second to denitrification. Simulating this would be a difficult task, probably out of the scope of the present article. Nevertheless, it would be useful if the authors would discuss this point**

Thank you for your comment. As you point out, the leaching flux of soil $NO_3^-$ in each layer is complex. Thus, we have added the following sentences in the manuscript (P11, L191-198 in revised manuscript).

The linear variation in the leaching flux and $\Delta^{17}O$ values of soil $NO_3^-$ used in the simulated calculations (Fig. 1) is just one of many possible variations in forested catchments. It is impossible to determine whether the linear variation was realistic or not until the downward water flux, along with the concentration and $\Delta^{17}O$ value of $NO_3^-$, was determined for each soil layer. However, the simultaneous observations of the oxygen isotopes of soil $NO_3^-$ and stream $NO_3^-$ (Hattori et al., 2019; Osaka et al., 2010; Nakagawa et al., 2018; Rose, 2014) implied that the approximation of the $\Delta^{17}O$ values of soil $NO_3^-$ to that of the stream $NO_3^-$ (Fig. 2b) was unrealistic.

**and especially if they could make recommendations on how to sample nitrate from the soil for $\Delta^{17}O$ determination: zero-tension lysimetry, tension lysimetry, centrifugation, extraction? I'm not sure if clear answers can be given with the present knowledge of soil nitrate transformations, but at least the question would deserve to be raised.**

Thank you for your comment. We have recommended the sampling method of soil nitrate in the manuscript (P11, L199-202 in revised manuscript).

If we estimate the downward water flux at each soil layer, with the $NO_3^-$ concentration and $\Delta^{17}O$ value of $NO_3^-$ in each soil layer using, e.g., a tension-free lysimeter (Inoue et al., 2021), we could estimate the vertical change in the leaching flux of $NO_3^-$ for each soil layer along with the $\Delta^{17}O$ of soil $NO_3^-$.

**L. 3: the word "eluted" is rather used for the what is done on purpose in the lab. In this case, for the process observed in the nature, a better choice would probably be "leached".**

Thank you for your suggestion. We changed the "eluted" to "leached" in the revised manuscript (P2/L4).

**Line 6: instead "nitrate metabolized", it would be better to write "nitrate that is consumed", first because as soon as it is consumed, it is no longer nitrate, and second because "metabolized" is rather used to indicate that it is incorporated into organic matter, which is not the case for the denitrification.**

Thank you for your suggestion. We changed the "nitrate metabolized" to "nitrate consumed" in the revised manuscript (P2/L6).

**L. 24: on the same idea: "consumption" instead of "metabolic".**

Thank you for your suggestion. We changed the "metabolic" to "consumption" in the revised manuscript (P3/L23).

**L. 27: "is negligible" is too general, better add "often".**

Thank you for your suggestion. We revised this in the revised manuscript (P3/L26).

**L. 28: "by order of magnitude": do you mean "one" order?**

We revised the sentence to "the GNR often exceeds the net nitrification rate by several orders of magnitude." (P3/L28-29)

**L. 21-29: very long sentence.**

We revised the sentence in the revised manuscript (P2-3/L21-29).

The net nitrification rate can be estimated from an increase in $NO_3^-$ concentration during a certain period. However, the gross nitrification rate (GNR) (net nitrification rate + consumption rate of $NO_3^-$ (e.g., that assimilated by plants or decomposed through denitrification)), reflects the internal N cycling better than the net nitrification rate (Bengtsson et al., 2003), especially in forested ecosystems. Although the net nitrification rate is often negligible (Stark and Hart, 1997), the consumption rate is significant in forested ecosystems, such that the GNR often exceeds the net nitrification rate by several orders of magnitude (Verchot et al., 2001).

**L. 31: it would be useful to explain shortly that the Δ anomaly is based on the δ of both $^{17}O$ and $^{18}O$ and that it is purposely defined so as to make it independent of mass-dependent fractionation.**

We added the information in the revised manuscript (P3-4/L41-44).

This is because possible variations in the $\delta^{17}O$ and $\delta^{18}O$ values during the processes of

biogeochemical isotope fractionation follow the relation of $\delta^{17}O \approx 0.5\ \delta^{18}O$, which cancels out the variations in the $\Delta^{17}O$ value.

**L. 31: in my opinion, "conservative" would be better than "conserved" (because it tends to be conserved but it is not always perfectly conserved).**

Thank you for your suggestion. We changed the "conserved" to "conservative" in the revised manuscript (P3/L31).

**L. 33: it seems strange to write "REmineralized" when it may be mineralized for the first time after centuries of N staying in the soil in the organic matter.**

We revised "remineralized nitrate ($NO_3^-{}_{re}$)" to "biologically produced nitrate ($NO_3^-{}_{bio}$)" in the revised manuscript (P3/L32-33).

**L. 34-38, 53-57: long sentences.**

We revised the sentences in the revised manuscript (P3/L34-39; P4/L57-60).

The $NO_3^-{}_{bio}$ always shows the $\Delta^{17}O$ value close to 0 ‰ because its oxygen atoms are derived from either terrestrial $O_2$ or $H_2O$ through nitrification. Contrarily, the $NO_3^-{}_{atm}$ always displays an anomalous enrichment in $^{17}O$ with $\Delta^{17}O$ value being approximately $+26 \pm 3$ ‰ in Japan (Tsunogai et al., 2010, 2016; Ding et al., 2022, 2023) because of oxygen transfers from atmospheric ozone (Michalski et al., 2003; Nelson et al., 2018).

Using the deposition flux of $NO_3^-{}_{atm}$ into the catchment and the leaching flux of unprocessed $NO_3^-{}_{atm}$ and $NO_3^-{}_{bio}$ from streams, the GNR in a forested catchment was estimated similarly to the estimation for water environments (Fang et al., 2015).

**L. 76: in this equation, some processes are denoted as subscript of $NO_3^-$ (like deposition) while others are denoted for themselves (like GNR). GNR and GDR are usually expressed as a nitrogen rather than as a nitrate flux. As it is written, the equation lets it open. It would be better to explicitly express all rates either as N or as $NO_3^-$.**

Thank you for your suggestion. Our simulation was done for the forested catchment reported by Hattori et al. (2019). Thus, the symbols used in the manuscript were in accordance with Hattori et al. (2019) as well. We mentioned this in the revised manuscript (P8/L144-145).

All the symbols (e.g., GNR) used here were consistent with those of Hattori et al. (2019).

**L. 74-80, 85-90: it is not clear why the word "each" is always used for the catchments (not only here, in general in the text).**

We removed the "each" in the revised manuscript.

**L. 112-113: repeated usage of the word "limited".**

We revised this in the revised manuscript (P7/L113).

**L. 116: which one of the $\Delta^{17}O$ is this? Or is it the difference?**

The $\Delta^{17}O$ denotes the $\Delta^{17}O$ of stream nitrate. We added this in the revised manuscript (P7/L118).

**L. 127-128: fine roots would be much more relevant than the total root biomass (with coarse roots obviously overrepresented close to the stem and thus close to the surface).**

Thank you for your suggestion. We added the information in the revised manuscript (P8/L128-131).

As most fine roots and root biomass are concentrated in the top 10 cm of the soil in forested catchments (Jackson et al., 1996; Li et al., 2020), most uptake reactions should occur in that top 10 cm of soil.

**L. 146-148: it may be useful to explain this as a gradual uptake (consumption) of nitrate as water moves down the profile.**

Thank you for your suggestion. We added the information in the revised manuscript (P9/L151-152).

This simulated the gradual net consumption of $NO_3^-$ in accordance with water flow in forested soils.

**L. 152-160: these assumptions are obviously simplifications compared to real measurements, but they make sense for the demonstration. It would be interesting to test also the assumption of nitrate fluxes increasing with depth because of a positive net nitrification.**

Thank you for your suggestion. We have simulated the positive net nitrification in the soil layers above.

**L. 175-179: long sentence.**

We revised the sentence in the revised manuscript (P11/L195-198).

However, the simultaneous observations of the oxygen isotopes of soil $NO_3^-$ and stream $NO_3^-$ (Hattori et al., 2019; Osaka et al., 2010; Nakagawa et al., 2018; Rose, 2014) implied that the approximation of the $\Delta^{17}O$ values of soil $NO_3^-$ to that of the stream $NO_3^-$ (Fig. 2b) was unrealistic.

**L. 220: as written, it is like anonymous reviewers would be named, which does not make sense.**

We revised the sentence in the revised manuscript (P13/L228-229).

We thank Dr. Joel Bostic, Dr. Lucy Rose and other two anonymous referees, for their valuable remarks on an earlier version of this paper.

**Fig. 1, fig. 2: the soil does not float above water and therefore "soil layers" and "water layer" should rather be marked "unsaturated soil layers" and either "water-saturated soil layer" or "seepage water" (as these two are considered to exhibit the same flux).**

We revised the "soil layers" and "water layer" to "unsaturated soil layers" and "seepage water" in the revised manuscript, respectively.

[Figure]

**Reference**

Hattori, S., Nuñez Palma, Y., Itoh, Y., Kawasaki, M., Fujihara, Y., Takase, K., and Yoshida, N.: Isotopic evidence for seasonality of microbial internal nitrogen cycles in a temperate forested catchment with heavy snowfall, Science of the Total Environment, 690, 290–299, https://doi.org/10.1016/j.scitotenv.2019.06.507, 2019.

Inoue, T., Nakagawa, F., Shibata, H., and Tsunogai, U.: Vertical Changes in the Flux of Atmospheric Nitrate From a Forest Canopy to the Surface Soil Based on $\Delta^{17}O$ Values, Journal of Geophysical Research: Biogeosciences, 126, 1–18, https://doi.org/10.1029/2020JG005876, 2021.

Li, F. L., McCormack, M. L., Liu, X., Hu, H., Feng, D. F., and Bao, W. K.: Vertical fine-root distributions in five subalpine forest types shifts with soil properties across environmental gradients, Plant Soil, 456, 129–143, https://doi.org/10.1007/s11104-020-04706-x, 2020.

Nakagawa, F., Tsunogai, U., Obata, Y., Ando, K., Yamashita, N., Saito, T., Uchiyama, S., Morohashi, M. and Sase, H.: Export flux of unprocessed atmospheric nitrate from temperate forested catchments: A possible new index for nitrogen saturation, Biogeosciences, 15(22), 7025–7042, doi:10.5194/bg-15-7025-2018, 2018.

Osaka, K., Ohte, N., Koba, K., Yoshimizu, C., Katsuyama, M., Tani, M., Tayasu, I., and Nagata, T.: Hydrological influences on spatiotemporal variations of $\delta^{15}N$ and $\delta^{18}O$ of nitrate in a forested headwater catchment in central Japan: Denitrification plays a critical role in groundwater , Journal of Geophysical Research: Biogeosciences, 115, n/a-n/a, https://doi.org/10.1029/2009jg000977, 2010.

Rose, L. A.: Assessing the nitrogen saturation status of appalachian forests using stable isotopes of nitrate [PhD thesis, University of Pittsburgh]. Retrieved from http://d-scholarship.pitt.edu/22783/1/LRose_ETD_081914_revised1.pdf.

**Response to the referee #2:**

**The authors assume in Equation (4),**
**Δ17O(NO3)uptake = Δ17O(NO3)denitrification = Δ17O(NO3)stream.**
**However, this assumption is not necessarily correct. It requires the assumption that nitrates deposited from the atmosphere are first diluted by nitrification (increasing nitrate amount with decreasing D17O) and then (i.e., "afterward"), reduced in nitrate amount without changing D17O by uptake and/or denitrification.**

Who assumed Eq. (4) were the authors of the papers in which Eq. (6) had been used to estimate GNR, such as Fang et al. (2015), Hattori et al. (2019), and Huang et al. (2020). While none of the authors clarified that they had assumed Eq. (4) in their papers, Eq. (4) should be needed to derive Eq. (6) from Eqs. (2) and (3). In addition, we also presented that this assumption (Eq. (4)) is not necessarily correct. In short, you have the same opinion with us at least on this point.

**Another reverse possibility could be that atmospheric nitrates are reduced in quantity through uptake and/or denitrification without changing D17O, and then nitrates are added through nitrification (by decreasing D17O). In this assumption, one could hypothesize:**
**D17O_uptake = D17O_denitrification = D17O_atm   (A1),**
**and calculate GNR as follows:**
**GNR = NO3_st × (D17O_atm – D17O_st) / D17O_st   (A2).**
**To compare using Equation (4) versus Equations A1 and A2, let's assume a system where 100 nitrates (assuming D17O is 24‰) are initially deposited. In this case, when suppose the stream water nitrate is also 100 but with D17O decreased to 3‰. Using the same assumption as the authors (using Eq. 4 and 6), GNR is calculated as 700 using Equation (6) in the manuscript (GNR = 100 x (24-3)/3 = 700). However, assuming A1 and A2, GNR can be calculated as 87.5 (GNR = 100 x (24-3)/24 = 87.5), which is an extremely lower result compared to another case. Yet, in both outcomes, the final stream water remains the same at 100 in nitrate amount and 3‰ in D17O of nitrate from the same starting point (100 of nitrate with D17O = 24‰). It is necessary to find a converging point by differentiation, and it can be understood that this is the "heterogeneous" method assumed by the authors in the manuscript with 10 soil layers. In the above-mentioned case, a GNR of ~208 will be the case when considering production and consumption occur simultaneously, as far as I calculated briefly (dividing layers > 1000).**

The equation A2 you wrote may be a typo. Under the assumption of A1, A2 should be:
GNR = NO3_st × (D17O_atm – D17O_st) / D17O_atm                     (RA2)
The equation A1 can be possible for forested catchments in which possible variations

in both the leaching flux and $\Delta^{17}O$ values of soil $NO_3^-$ were not determined for each soil layer. When we apply the equation RA2 to the forested catchment we used for the simulation (i.e. the forested catchment studied by Hattori et al., 2019), we obtain much smaller GNR of 2.4 kg of N ha$^{-1}$ y$^{-1}$ (GNR = 2.6 x (28-2.2)/28 = 2.4) compared to the GNR calculated by using Eq. (6) (83.6 kg of N ha$^{-1}$ y$^{-1}$; GNR = 7.0 x (28-2.2)/2.2 = 83.6) that had been used in the literatures (Fang et al., 2015; Hattori et al., 2019; Huang et al., 2020). Thus, you reached the same conclusion with us that the GNR estimated from Eq. (6) using the $\Delta^{17}O$ values of stream nitrate was, to some extent, an overestimate of the actual GNR.

**In reality, production and consumption occur simultaneously. Therefore, both cases may overestimate or underestimate GNR to an extreme.**

Both nitrification and consumption (uptake + denitrification) of $NO_3^-$ usually occur simultaneously in forested soil, as you pointed out. This is the reason we done a simulated calculation for the case shown in Figure 1 in the manuscript, in which both nitrification and consumption of $NO_3^-$ occur simultaneously in the soil.

**Thus, authors should consider this case considering equations A1 and A2, in addition to the case considered in this study.**

Thank you for your advice. The aim of this paper is to clarify that the GNR estimated by using Eq. (6) was not the only GNR that can be expected in each forested catchment. Rather, the GNR estimated by using Eq. (6) often overestimate actual GNR to some extent. We trust that the case shown in Figure 1 is sufficient to accomplish our aim shown above.

**Additionally, the authors have limited their verification of GNR calculation overestimation in their manuscript (underestimation in the case of A1 and A2 in this review report) to the soil profile. However, if pointing out such overestimation in GNR calculation methods, it would be better to also consider similar considerations for N cycling rates (e.g., GNR) calculated for lake systems, as advanced by the authors' group. Hasn't there been an overestimation for similar reasons in studies using nitrogen cycling rates in Lake systems, as shown in Tsunogai et al. (2011 and 2018) and other previous research? In lake and/or river studies, might they have calculated rates assuming that nitrates are added by nitrification (increasing the amount and decreasing D17O), and then the amount reduces by uptake and denitrification without changing D17O "only once" within each observation period unit (monthly or quarterly)? Wouldn't both assumptions based on Equation 4 and those similar to A1 and A2 be equally valid? Assuming simultaneous production and consumption as in lake mass balance calculations, converging to a single value might provide a more reliable N cycle rate. It should**

**also be pointed out that the authors' group's previous N cycling research may have been overestimated.**

Your claim on our studies applying the $\Delta^{17}O$ tracer to water environments is wrong. In case of the water environments, differ from forested catchments, the $\Delta^{17}O$ values of $NO_3^-$ were mostly homogeneous in the water column due to the active vertical mixing in the water column during cold seasons and storm events. Additionally, the homogeneity of the $\Delta^{17}O$ values had been verified through actual observation prior to calculating GNR (Tsunogai et al., 2011, 2018). Furthermore, the extent of heterogeneities of the $\Delta^{17}O$ values in the water column had been evaluated in calculating GNR etc., so that the calculated values of GNR were reported with the ranges of errors (Tsunogai et al., 2011, 2018). These are the essential differences between the past studies on the water environments and those on the forested catchments using Eq. (6) to estimate GNR.

**Especially, since Tsunogai et al. (2018) concluded that the nitrogen cycle rate was faster compared to 15N tracer experiments, which makes their study significant, it is important to consider the possibility of overestimation. Overall, this manuscript should consider and comment also on the case of their application for other systems like lake/river.**

Your understanding on Tsunogai et al. (2018) is wrong. Please note that the difference in the fluxes between the $\Delta^{17}O$ method and the $^{15}N$ tracer method estimated in a water environment by Tsunogai et al. (2018) was only 20 % on the annual base, while the difference in the forested catchment between the calculation methods was more than 500 %. In addition, Tsunogai et al. (2018) had discussed the reason for the difference (20 %) between the $\Delta^{17}O$ method and the $^{15}N$ tracer method in detail in the paper and concluded that the differences in the period of observation (instantaneous for the $^{15}N$ tracer method vs. long-range average for the $\Delta^{17}O$ method) were primarily responsible for the discrepancy so that the reason was essentially different from the discrepancy in the forested catchment. We don't see any merit in discussing the water environments again in this manuscript.

**Based on the above two major comments, here are some suggestions for the cases considered in this study:**
  **1. Consider that the case of Equation (4) may not always be correct.**

As we already explained, those who assumed Eq. (4) were the authors of the papers in which Eq. (6) had been used to estimate GNR, such as Fang et al. (2015), Hattori et al. (2019), and Huang et al. (2020). We presented that this assumption (Eq. (4)) is not necessarily correct, in lines from 66 to 68 and from 131 to 134 in the revised manuscript, so that you have the same opinion with us on this point.

**Consider also the case assuming Equations A1 and A2 provided in this review report.**

As we already explained, the aim of this paper is to clarify that the GNR estimated by using Eq. (6) was not the only GNR that can be expected in each forested catchment. Rather, the GNR estimated by using Eq. (6) often overestimate actual GNR to some extent. We trust that showing the case Figure 1 is sufficient to accomplish our aim.

2. **Instead of comments using other group's case as an example, verify the calculation process and resulting GNR in the more general system.**

As presented in the manuscript (L199-209 in the revised manuscript), our conclusion is that it is impossible to estimate reliable GNR in each ecosystem (e.g., forested catchments, lakes, glaciers) in general using $\Delta^{17}O$ as a tracer without measurement on the $\Delta^{17}O$ values of $NO_3^-$ actually consumed in each ecosystem. It is impossible to present the calculation process in the more general system without actual observation.

3. **Not only soil profile cases, but also consider possible changes for the other systems led by their research group (e.g., Tsunogai et al. 2011 Biogeos; Tsunogai et al. 2018 L&O).**

In case of the water environments, differ from forested catchments, the $\Delta^{17}O$ values of $NO_3^-$ were mostly homogeneous in the water column due to the active vertical mixing in the water column during cold seasons and storm events. Additionally, the homogeneity of the $\Delta^{17}O$ values had been verified through actual observation prior to calculating GNR (Tsunogai et al., 2011, 2018). Furthermore, the extent of heterogeneities of the $\Delta^{17}O$ values in the water column had been evaluated in calculating GNR etc., so that the calculated values of GNR were reported with the ranges of errors (Tsunogai et al., 2011, 2018). These are the essential differences between the past studies on the water environments and those on the forested catchments using Eq. (6) to estimate GNR. We added the new information to emphasized this in the revised manuscript (P7/L114-116)

Differ from water environments, vertical mixing of water/soil is difficult in forested soil, so the $\Delta^{17}O$ values of soil $NO_3^-$ are often heterogeneous.

**I also note that the current manuscript seems to criticize other groups' research, which may be due to language issues, so I want to avoid pointing out each by each in this review report. However, it might be worthwhile for the authors to reflect similar self-criticism on their group's previous nitrogen cycle research.**
**To be honest, the current manuscript feels like an incomplete consideration that**

**criticizes others' research one-sidedly. I also note that a similar modification of the calculation way for GNR based on D17O, considering both the production and consumption of nitrate simultaneously, has been already considered/published in another paper (Hattori et al. 2023).**

Because we found a problem in applying the $\Delta^{17}O$ method to forested catchments by using Eq. (6) to estimate GNR as Fang et al. (2015) did, we just ignored and did not estimate GNR in our subsequent manuscripts studying forested catchments using $\Delta^{17}O$ of $NO_3^-$ as a tracer, such as Nakagawa et al. (2018) and Ding et al. (2022, 2023). Because no one pointed out the problem to use Eq. (6) in calculating GNR in forested catchments, however, apparently overestimated GNR by using Eq. (6) became "normal" in the papers subsequent to Fang et al. (2015) (Hattori et al., 2019, Huang et al., 2020), which seems to have reduced reliability of the $\Delta^{17}O$ method. We trust this paper is worthy of publication in Biogeosciences to clarify the problem inherited in this method. Concerning to Hattori et al. (2023), please note that the first preprint of our paper was published on 12 Jan 2023 (https://bg.copernicus.org/preprints/bg-2022-236/). Because this was 5 months earlier than the submission of Hattori et al. (2023), who should "consider" must be the authors of Hattori et al. (2023).

**Title: It is better to replace "error" with "bias"?**

Thank you for your advice. We revised the "error" to "bias" in the revised manuscript.

**L149: Why 10 layers? If you consider fewer or more layers, do you expect any changes?**

Thank you for your questions. Dividing the forested soils into more layers can enhance the precision of the simulated GNR. By dividing the forested soils into 10, 20, 30, 50, 100, and 1000 layers, the simulated GNR was 13.0, 11.4, 11.0, 10.5, 10.3, and 10.1 kg of N ha$^{-1}$ y$^{-1}$, respectively. We added this information to the revised manuscript (P10/L171-174).

Even if the number of layers in the forested soils was increased to 20, 30, 50, 100, and 1000 to enhance the precision of the GNR simulated for the catchment with the heterogeneous soil, the GNR was 11.4, 11.0, 10.5, 10.3, and 10.1 kg of N ha$^{-1}$ y$^{-1}$, respectively.

**Reference**
Ding, W., Tsunogai, U., Nakagawa, F., Sambuichi, T., Sase, H., Morohashi, M., and Yotsuyanagi, H.: Tracing the source of nitrate in a forested stream showing elevated concentrations during storm events, Biogeosciences, 19, 3247–3261, https://doi.org/10.5194/bg-19-3247-2022, 2022.

Ding, W., Tsunogai, U., Nakagawa, F., Sambuichi, T., Chiwa, M., Kasahara, T., and Shinozuka, K.: Stable isotopic evidence for the excess leaching of unprocessed atmospheric nitrate from forested catchments under high nitrogen saturation, Biogeosciences, 20, 753–766, https://doi.org/10.5194/bg-20-753-2023, 2023.

Fang, Y., Koba, K., Makabe, A., Takahashi, C., Zhu, W., Hayashi, T., Hokari, A. A., Urakawa, R., Bai, E., Houlton, B. Z., Xi, D., Zhang, S., Matsushita, K., Tu, Y., Liu, D., Zhu, F., Wang, Z., Zhou, G., Chen, D., Makita, T., Toda, H., Liu, X., Chen, Q., Zhang, D., Li, Y. and Yoh, M.: Microbial denitrification dominates nitrate losses from forest ecosystems, Proc. Natl. Acad. Sci. U. S. A., 112(5), 1470–1474, doi:10.1073/pnas.1416776112, 2015.

Hattori, S., Nuñez Palma, Y., Itoh, Y., Kawasaki, M., Fujihara, Y., Takase, K. and Yoshida, N.: Isotopic evidence for seasonality of microbial internal nitrogen cycles in a temperate forested catchment with heavy snowfall, Sci. Total Environ., 690, 290–299, doi:10.1016/j.scitotenv.2019.06.507, 2019.

Huang, S., Wang, F., Elliott, E. M., Zhu, F., Zhu, W., Koba, K., Yu, Z., Hobbie, E. A., Michalski, G., Kang, R., Wang, A., Zhu, J., Fu, S. and Fang, Y.: Multiyear Measurements on $\Delta^{17}O$ of Stream Nitrate Indicate High Nitrate Production in a Temperate Forest, Environ. Sci. Technol., 54(7), 4231–4239, doi:10.1021/acs.est.9b07839, 2020.

Hattori, S., Li, Z., Yoshida, N., and Takeuchi, N.: Isotopic Evidence for Microbial Nitrogen Cycling in a Glacier Interior of High-Mountain Asia, Environ. Sci. Technol., 57, 15026–15036, https://doi.org/10.1021/acs.est.3c04757, 2023.

Nakagawa, F., Tsunogai, U., Obata, Y., Ando, K., Yamashita, N., Saito, T., Uchiyama, S., Morohashi, M. and Sase, H.: Export flux of unprocessed atmospheric nitrate from temperate forested catchments: A possible new index for nitrogen saturation, Biogeosciences, 15(22), 7025–7042, doi:10.5194/bg-15-7025-2018, 2018.

Tsunogai, U., Daita, S., Komatsu, D. D., Nakagawa, F. and Tanaka, A.: Quantifying nitrate dynamics in an oligotrophic lake using $\Delta^{17}O$, Biogeosciences, 8(3), 687–702, doi:10.5194/bg-8-687-2011, 2011.

Tsunogai, U., Miyauchi, T., Ohyama, T., Komatsu, D. D., Ito, M. and Nakagawa, F.: Quantifying nitrate dynamics in a mesotrophic lake using triple oxygen isotopes as tracers, Limnol. Oceanogr., 63, S458–S476, doi:10.1002/lno.10775, 2018.

---

## Author Response (AR2)

Aug 21, 2024

Dr. Frank Hagedorn
Editor of Biogeosciences

Title: Bias in calculating gross nitrification rates in forested catchments using the triple oxygen isotopic composition (Δ17O) of stream nitrate
Authors: Weitian Ding et al.
MS No.: egusphere-2023-2753

Dear Dr. Frank Hagedorn:

Thank you very much for handling our manuscript. We would like to thank the referees as well for the constructive comments on our manuscript. We have carefully studied the comments and revised the manuscript accordingly. We include below point-by-point responses to the comments, and detailed descriptions of the modifications we made to the manuscript. Besides, we also uploaded the revised manuscript in MS Word, in which all the revisions from BGD version were recorded. We hope that with these changes you will find our revised manuscript appropriate for publication in your journal.

Sincerely yours,
Weitian Ding
Postdoctoral researcher
Graduate School of Environmental Studies,
Nagoya University
Furo-cho, Chikusa-ku, Nagoya,
464-8601, JAPAN
Phone: +81-70-4436-3157
E-mail: dwt530754556@gmail.com
Cc: Drs. Urumu Tsunogai and Fumiko Nakagawa

**Response to the referee #1:**

**My recommandations made for the previous version of the manuscript were largely applied. The suggestion of trying a different profile of nitrate with soil depth was also realised, but only reported in the authors' answers and not in the article itself. The idea goes into the conclusion, but the abstract still only reports on the one simulation (verb "was" in its last sentence). The abstract therefore leaves the question open how generalisable the conclusion is. And essentially it would be.**

Thank you for your comment. We added the simulated calculation in the manuscript as follows (P2, L6-9; P11, L192-204 in revised manuscript).

The GNR estimated from the $\Delta^{17}O$ value of stream nitrate was significantly higher than the GNRs in our simulated calculations for a forested catchment where the soil nitrate had $\Delta^{17}O$ values higher than those the stream nitrate.

Furthermore, even if we assumed non-linear variation for the leaching flux of soil $NO_3^-$, in which the leaching flux of soil $NO_3^-$ increased with soil depth from layers 1 to 5 with an increasing rate of 0.44 kg of N $ha^{-1}$ $y^{-1}$ $layer^{-1}$, while the leaching flux decreased with soil depth from layers 6 to 10 with a decreasing rate of 1.32 kg of N $ha^{-1}$ $y^{-1}$ $layer^{-1}$ (Table S3), the newly estimated total GNR (19.1 kg of N $ha^{-1}$ $y^{-1}$) was still comparable with that estimated for the forested catchment with the heterogeneous soil shown by Figure 1 (13.0 kg of N $ha^{-1}$ $y^{-1}$). As a result, we concluded that the differences in the $\Delta^{17}O$ values of the soil $NO_3^-$ consumed in a forested catchment from that of stream $NO_3^-$ resulted in a significant deviation in the GNR estimated using Eq. 6 from the actual GNR. In addition, the most important parameter to determine GNR was the $\Delta^{17}O$ values of $NO_3^-$ consumed in soil layers. That is, the other parameters such as the number of layers and the vertical changes in the leaching flux of soil $NO_3^-$ had little impact on total GNR.

**L. 10: could be simplified to "with depth" (dropping "an increase in").**

Thank you for your comment. To improve the flow of abstract, we removed the sentence "The $\Delta^{17}O$ values of the soil nitrate decreased with an increase in depth to that of the stream nitrate at the bottom" from the abstract in the revised manuscript.

**L. 19-20: "forest ecosystems" could be replaced by "soils" (first because this is not only valid for forests, second because it avoids to repeat too many times "forest ecosystems").**

Thank you for your comment. We revised this in the revised manuscript as follows (P2,

L15 in the revised manuscript).

Nitrate ($NO_3^-$) is an important nitrogen nutrient for primary production in soils.

**L. 24: a parenthesis inside another parenthesis could be avoided simply by putting a comma between both parts.**

Thank you for your comment. We revised this in the revised manuscript as follows (P2, L19-23 in the revised manuscript).

However, the gross nitrification rate (GNR), which includes the net nitrification rate plus the consumption rate of $NO_3^-$ (e.g., through plant assimilation or denitrification), reflects the internal N cycling better than the net nitrification rate (Bengtsson et al., 2003), especially in forested ecosystems.

**L. 31: the comma before "as a" does not seem justified.**

Thank you for your comment. We removed the comma in the revised manuscript as follows (P3, L28 in the revised manuscript).

Recent studies have successfully estimated the GNR in aquatic environments, such as lakes, using the $\Delta^{17}O$ values of $NO_3^-$ as a conservative tracer to determine the mixing ratio between atmospheric nitrate ($NO_3^-{}_{atm}$) and biologically produced nitrate ($NO_3^-{}_{bio}$) (Tsunogai et al., 2011, 2018).

**L. 52: "also" should be introduced between "this is" and "because" (as it is just on more reason and not the single reason why the method is applicable).**

Thank you for your comment. We revised the sentence in the revised manuscript as follows (P4, L49-52 in the revised manuscript).

This approach works because the $NO_3^-{}_{atm}/NO_3^-{}_{total}$ ratios are homogeneous in the water column due to the active vertical mixing; thus, we can constrain the $NO_3^-{}_{atm}/NO_3^-{}_{total}$ ratios of $NO_3^-$ consumed in the lake water column (Tsunogai et al., 2011, 2018).

**L. 55: "applications" could be plural.**

Thank you for your comment. We revised the sentence in the revised manuscript as follows (P4, L53-55 in the revised manuscript).

In addition to applications in water environments, the $\Delta^{17}O$ method has been applied to forested catchments to determine GNR (Fang et al., 2015; Hattori et al., 2019; Huang

et al., 2020).

**L. 115: using the verb "differ" does not seem correct here. Please revise the sentence, reconsidering also the word "difficult" (something like "very limited" may be better).**

Thank you for your comment. We revised the sentence in the revised manuscript as follows (P7, L113-115 in the revised manuscript).

Different from water environments, vertical mixing of water/soil is limited in forested soil, so the $\Delta^{17}O$ values of soil $NO_3^-$ are often heterogeneous.

**L. 175 ff: this sentence is not well formed: as I understand it, one subject ("studies") would have two verbs ("estimated" and "were"). Please revise.**

Thank you for your comment. We revised the sentence in the revised manuscript as follows (P10, L181-184 in the revised manuscript).

This result allows us to further verify that past studies estimating GNR using Eq. 6 implicitly approximated that $\Delta^{17}O$ values of soil $NO_3^-$ consumed in forested catchments were homogeneous and always equal to those of stream $NO_3^-$.

**L. 206: the last sentence of the section repeats facts already described. My suggestion is to delete it and go directly to the conclusion.**

Thank you for your comment. We deleted the sentences in the revised manuscript.

**L. 229: there should be no comma after "referees".**

Thank you for your comment. We revised this in the revised manuscript as follows (P13, L241-242 in the revised manuscript).

We thank Dr. Joel Bostic, Dr. Lucy Rose and other two anonymous referees for their valuable remarks on an earlier version of this paper.

**Response to the referee #3:**

**The paper provides a set of model calculations estimating the GNR of a catchment, for the community to consider the vertical profiles of D17O in soil layers. The results presented, however, are way too vague and handwaving, particularly the vertical profiles of D17O and nitrate flux taken. I suggest the authors reconsider the validity of the model and assumptions and provide the needed justification.**

**1. GNR is largely affected by the vertical gradient of D17O in the soil. The main data the authors quoted are from Hattori et al. (2019), who provided limited information on the soil nitrate at depths. The authors should analyze that data in detail and compare to the gradient taken in the current work. Same for nitrate concentration.**

Thank you for the comment. Our simulation was done for the forested catchment reported by Hattori et al. (2019), as presented in the manuscript. While the $\Delta^{17}O$ value of $NO_3^-{}_{atm}$ was +28.0 ‰, the $\Delta^{17}O$ value of $NO_3^-{}_{stream}$ (nitrate leaching from the catchment) was +2.2 ‰ on the average in the catchment. In addition, they found a decreasing trend in the $\Delta^{17}O$ values of soil $NO_3^-$ with depth throughout their observation. Specifically, the measured mean $\Delta^{17}O$ values (average values of summer and winter season) of the soil $NO_3^-$ was +17, +4, +3, and +5 ‰ at depths of 0, 25, 55, and 90 cm from the soil surface, respectively. Similar decreasing trend had been found in the other forested catchment as well (Rose, 2014). These are the reasons we used the $\Delta^{17}O$ values of soil $NO_3^-$ showing decreasing trend with soil depth in our original simulations (linear variation), shown by Figures 1b and 2b in the manuscript.

While Hattori et al. (2019) reported the concentration of soil $NO_3^-$ for each layer showing little vertical variation in the forested catchment, they didn't measure the water flux in the catchment. Thus, it is difficult to constrain the vertical changes in the leaching flux of soil $NO_3^-$ from each layer in the forested catchment. Still, the deposition flux of $NO_3^-$ was 7.0 kg of N ha$^{-1}$ y$^{-1}$ and the final leaching flux of $NO_3^-$ via stream was estimated to be 2.6 kg of N ha$^{-1}$ y$^{-1}$ in the forested catchment (Hattori et al., 2019). In addition, the water flux always showed gradual decreasing trend with depth in various forested catchments (e.g., Christiansen et al., 2006). Thus, we used the linear decreasing variation in the leaching flux of soil $NO_3^-$ in our simulations, shown by Figures 1c and 2c in the manuscript. Similar decreasing trend in the leaching flux of soil $NO_3^-$ had been found in the other forested catchments as well (Callesen et al., 1999; Inoue et al., 2021). None of the vertical profiles of $\Delta^{17}O$ and leaching flux of soil $NO_3^-$ adopted in our model were "handwaving".

In response to your comment, we made a new simulated calculation in which the forested soil layers were divided vertically into 5 layers to increase the vertical gradient in the $\Delta^{17}O$ values between the layers (Table R1). While GNR was increased to 29.6 kg of N ha$^{-1}$ y$^{-1}$ from the original (13.0 kg of N ha$^{-1}$ y$^{-1}$), it was still significantly smaller

than the GNR calculated by using Eq.6 (83.6 kg of N ha$^{-1}$ y$^{-1}$). This additional simulated calculation also supports our conclusion that the GNR estimated from the $\Delta^{17}O$ value of stream nitrate in forested catchments can be an overestimate of the actual GNR.

**Table R1.** $\Delta^{17}O$ values of $NO_3^-$, leaching flux of $NO_3^-$, total consumption rate of $NO_3^-$ (GDR + uptake), and GNR in the simulated forested soil where the distribution of $\Delta^{17}O$ values of $NO_3^-$ is heterogeneous with the values in accordance with the measured $\Delta^{17}O$ mean values of soil $NO_3^-$ at different depth as reported by Hattori et al. (2019).

| Depth layer | $\Delta^{17}O$ ‰ | $NO_3^-$ flux | GDR +uptake | GNR |
|---|---|---|---|---|
| | | kg of N ha$^{-1}$ y$^{-1}$ layer$^{-1}$ | | |
| 0 | 28 | 7.0 | 0.0 | 0.0 |
| 1 | 17 | 6.1 | 5.4 | 4.5 |
| 2 | 4 | 5.2 | 20.8 | 19.9 |
| 3 | 3 | 4.4 | 2.6 | 1.7 |
| 4 | 5 | 3.5 | -0.9 | -1.7 |
| 5 | 2 | 2.6 | 6.1 | 5.2 |
| 6 | 2 | 2.6 | 0 | 0 |
| Total | | | 34.0 | 29.6 |

We added the information about the assumed $NO_3^-$ leaching flux to the revised manuscript as follows (P9-10, L160-170; P11, L192-204 in the revised manuscript).

To estimate GNR in each layer, both the $\Delta^{17}O$ value and the $NO_3^-$ leaching flux in soil are required. While Hattori et al. (2019) reported soil $NO_3^-$ concentrations for each layer, indicating little vertical variation within the forested catchment, they did not measure the catchment water flux. Consequently, it is difficult to constrain the $NO_3^-$ leaching flux for each layer of forest soil. Nevertheless, $NO_3^-_{deposition}$ was 7.0 kg N ha$^{-1}$ y$^{-1}$ and $NO_3^-_{leaching}$ was 2.6 kg N ha$^{-1}$ y$^{-1}$ in the catchment (Hattori et al., 2019). Additionally, because water fluxes decrease gradually with depth in various forest settings (e.g., Christiansen et al., 2006), we assumed a gradual decrease in $NO_3^-$, leaching flux from 7.0 to 2.6 kg N ha$^{-1}$ y$^{-1}$ at a rate of $-0.44$ kg N ha$^{-1}$ y$^{-1}$ per layer (Figs. 1c and 2c). Similar trends in the $NO_3^-$ leaching flux of soil have been observed in other forested catchments (Callesen et al., 1999; Inoue et al., 2021).

Furthermore, even if we assumed non-linear variation for the leaching flux of soil $NO_3^-$, in which the leaching flux of soil $NO_3^-$ increased with soil depth from layers 1 to 5 with an increasing rate of 0.44 kg of N ha$^{-1}$ y$^{-1}$ layer$^{-1}$, while the leaching flux decreased with soil depth from layers 6 to 10 with a decreasing rate of 1.32 kg of N ha$^{-1}$ y$^{-1}$ layer$^{-1}$ (Table S3), the newly estimated total GNR (19.1 kg of N ha$^{-1}$ y$^{-1}$) was still comparable

with that estimated for the forested catchment with the heterogeneous soil shown by Figure 1 (13.0 kg of N ha$^{-1}$ y$^{-1}$). As a result, we concluded that the differences in the $\Delta^{17}O$ values of the soil $NO_3^-$ consumed in a forested catchment from that of stream $NO_3^-$ resulted in a significant deviation in the GNR estimated using Eq. 6 from the actual GNR. In addition, the most important parameter to determine GNR was the $\Delta^{17}O$ values of $NO_3^-$ consumed in soil layers. That is, the other parameters such as the number of layers and the vertical changes in the leaching flux of soil $NO_3^-$ had little impact on total GNR.

**2. If I understand the model correctly, the authors implicitly assumed steady-state that the soil nitrate profile is nonvarying, inconsistent with variable profiles seen by Hattori et al.**

Thank you for your comment. Please note that the y-axes in our simulated models were layers, not depths. While the $\Delta^{17}O$ values of $NO_3^-$ always showed decreasing trends with depths irrespective to the seasons, $\Delta^{17}O$ values of soil $NO_3^-$ showed significant temporal variation at each depth (Hattori et al., 2019). This was the reason why the layers were adopted for the y-axes in our models, instead of depths.

As a result, the specific depth of each layer varies over time. In addition, the relation between depth and layer is not always linear. The temporal variation found in the vertical distributions of $\Delta^{17}O$ values can be explained by this model as well without contradiction because the $\Delta^{17}O$ values of soil $NO_3^-$, while showing large temporal variation at each depth, always showed decreasing trend with depth throughout their observation (Hattori et al., 2019).

On the other hand, those who used Eq.6, such as Fang et al. (2015), Hattori et al. (2019), and Huang et al. (2020), implicitly assumed the $\Delta^{17}O$ values of $NO_3^-$ in the soil, where GDR and uptake occurred, to be "steady state" at the $\Delta^{17}O$ value of stream $NO_3^-$ (+2.2 ‰), while actual $\Delta^{17}O$ values of soil $NO_3^-$ were variable temporally and generally higher than +2.2 ‰, as you point out. This was the reason we concluded that GNR estimated by using Eq.6 was highly inaccurate and submitted this manuscript.

We added this information to the revised manuscript as follows (P9, L149-159 in the revised manuscript).

Note that the y-axes in the models were layers, not depths (Tables S1, S2, and S3). While the $\Delta^{17}O$ values of soil $NO_3^-$ always showed decreasing trends with depths irrespective to the seasons, $\Delta^{17}O$ values of soil $NO_3^-$ showed significant temporal variation at each depth (Hattori et al., 2019). This was the reason why the layers were adopted for the y-axes in our models, instead of depths. As a result, the specific depth of each layer varies over time. In addition, the relation between depth and layer is not always linear. The temporal variation found in the vertical distributions of $\Delta^{17}O$ values in the forested catchment (Hattori et al., 2019) can be explained by our model as well

without contradiction because the $\Delta^{17}O$ values of soil $NO_3^-$, while showing large temporal variation at each depth, always showed decreasing trend with depth throughout their observation (Hattori et al., 2019).

**3. No discussion on the nitrate resident time in the soil column. To have the model working requires GDR/uptake/GNR time scale less than the transport/diffusion time in each layer.**

Thank you for the comment. As already discussed in past studies (Tsunogai et al., 2011; 2018), the GNR can be calculated from the isotopic mass balance (Eq. (3); $NO_3^-{}_{deposition} \times \Delta^{17}O(NO_3^-)_{atm} + GNR \times \Delta^{17}O(NO_3^-)_{nitrification} = NO_3^-{}_{leaching} \times \Delta^{17}O(NO_3^-)_{stream} + NO_3^-{}_{uptake} \times \Delta^{17}O(NO_3^-)_{uptake} + GDR \times \Delta^{17}O(NO_3^-)_{denitrification}$), so that the parameter of residence time in the soil column is not necessary for calculating GNR. This is the merit to determine $\Delta^{17}O$ of $NO_3^-$ for those we can constrain the values of $\Delta^{17}O(NO_3^-)_{uptake}$ and $\Delta^{17}O(NO_3^-)_{denitrification}$.

**4. The obtained GNR increases with depths (Figure 1e). Nitrification is minimal at low oxygen conditions. How significant is the nitrification in deep soils?**

Because Hattori et al. (2019) didn't report the water flux for each soil layer, the linear variation in the leaching flux and $\Delta^{17}O$ values of soil $NO_3^-$ used in the simulated calculations (Figure 1) is just one of the many possible vertical variations in forested catchments. Thus, the calculated vertical distribution of GNR was also one of many possible distributions.

Nevertheless, it is not surprising that the nitrification rates in deeper layers are comparable to those in surface layers in the forested catchments with high precipitation, because soil water is generally enriched in $O_2$ throughout the soil layers in such high precipitation area. For example, using the $^{15}N$-pool dilution technique in three different forest soil layers (organic layer (surface layers), 0–10 cm depth, 10–40 cm depth) at Hoglwald Forest (Bavaria, Germany), Matejek et al. (2010) found active gross nitrification rate, up to $1600 \pm 700$ μmol N m$^{-2}$ d$^{-1}$ layer$^{-1}$ at the depths of 10–40 cm, while $600 \pm 200$ at the depths of 0–10 cm and $2000 \pm 300$ μmol N m$^{-2}$ d$^{-1}$ layer$^{-1}$ at the organic layer. Such active nitrification in deep layers in the literatures also implied that the vertical distribution of GNR estimated by using Eq. 6 (Figure 2e) was unrealistic, in which GNR should be concentrated only at the surface soil layers.

**1. I believe NO3- flux reported is the flux at the layer boundary and GDR+uptake/GNR are in the layer. And so, the GDR+uptake/GNR unit should be kgN/ha/y/layer. Please clarify.**

Thank you for your advice. We revised the units in Figures 1, 2, Tables S1, S2 and S3.

**2. Figures 1 and 2 captions: I believe (b) and (c) are assumed, not simulated.**

Thank you for your comment. We revised these in the revised manuscript.

**3. Line 115: Different from …**

Thank you for your comment. We revised the sentence in the revised manuscript as follows (P7, L113-115 in the revised manuscript).

Different from water environments, vertical mixing of water/soil is limited in forested soil, so the $\Delta^{17}O$ values of soil $NO_3^-$ are often heterogeneous.

**4. Line 173-175: A best is to do integral, not summation and to play with different D17O and nitrate flux gradients on the GNR.**

Thank you for your comment. Compared to the summation, integration can enhance the precision of the simulated GNR. By dividing the forested soils into 10 layers, the result of integration for the simulated GNR for the forested catchment with the profile shown by Figure 1 was 11.2 kg of N ha$^{-1}$ y$^{-1}$ ($\int_0^{10} \frac{31.52-2.27x}{28-2.58x} - 0.44 dx$).

We added this information to the revised manuscript as follows (P11, L190-192 in the revised manuscript).

Moreover, when we changed the calculation method from stepwise summation to integration, the estimated GNR was 11.2 kg N ha$^{-1}$ y$^{-1}$.

**Reference**

Callesen, I., Raulund-Rasmussen, K., Gundersen, P., and Stryhn, H.: Nitrate concentrations in soil solutions below Danish forests, Forest Ecology and Management, 114, 71–82, https://doi.org/10.1016/S0378-1127(98)00382-X, 1999.

Christiansen, J. R., Elberling, B., and Jansson, P.-E.: Modelling water balance and nitrate leaching in temperate Norway spruce and beech forests located on the same soil type with the CoupModel, Forest Ecology and Management, 237, 545–556, https://doi.org/10.1016/j.foreco.2006.09.090, 2006.

Fang, Y., Koba, K., Makabe, A., Takahashi, C., Zhu, W., Hayashi, T., Hokari, A. A., Urakawa, R., Bai, E., Houlton, B. Z., Xi, D., Zhang, S., Matsushita, K., Tu, Y., Liu, D., Zhu, F., Wang, Z., Zhou, G., Chen, D., Makita, T., Toda, H., Liu, X., Chen, Q., Zhang, D., Li, Y. and Yoh, M.: Microbial denitrification dominates nitrate losses from forest ecosystems, Proc. Natl. Acad. Sci. U. S. A., 112(5), 1470–1474, doi:10.1073/pnas.1416776112, 2015.

Hattori, S., Nuñez Palma, Y., Itoh, Y., Kawasaki, M., Fujihara, Y., Takase, K. and Yoshida, N.: Isotopic evidence for seasonality of microbial internal nitrogen cycles in a temperate forested catchment with heavy snowfall, Sci. Total Environ., 690, 290–299, doi:10.1016/j.scitotenv.2019.06.507, 2019.

Huang, S., Wang, F., Elliott, E. M., Zhu, F., Zhu, W., Koba, K., Yu, Z., Hobbie, E. A., Michalski, G., Kang, R., Wang, A., Zhu, J., Fu, S. and Fang, Y.: Multiyear Measurements on $\Delta^{17}$O of Stream Nitrate Indicate High Nitrate Production in a Temperate Forest, Environ. Sci. Technol., 54(7), 4231–4239, doi:10.1021/acs.est.9b07839, 2020.

Inoue, T., Nakagawa, F., Shibata, H. and Tsunogai, U.: Vertical Changes in the Flux of Atmospheric Nitrate From a Forest Canopy to the Surface Soil Based on $\Delta^{17}$O Values, J. Geophys. Res. Biogeosciences, 126(4), 1–18, doi:10.1029/2020JG005876, 2021.

Matejek, B., Huber, C., Dannenmann, M., Kohlpaintner, M., Gasche, R., Göttlein, A., and Papen, H.: Microbial nitrogen-turnover processes within the soil profile of a nitrogen-saturated spruce forest and their relation to the small-scale pattern of seepage-water nitrate, Journal of Plant Nutrition and Soil Science, 173, 224–236, https://doi.org/10.1002/jpln.200800226, 2010.

Tsunogai, U., Daita, S., Komatsu, D. D., Nakagawa, F. and Tanaka, A.: Quantifying nitrate dynamics in an oligotrophic lake using $\Delta^{17}$O, Biogeosciences, 8(3), 687–702, doi:10.5194/bg-8-687-2011, 2011.

Tsunogai, U., Miyauchi, T., Ohyama, T., Komatsu, D. D., Ito, M. and Nakagawa, F.: Quantifying nitrate dynamics in a mesotrophic lake using triple oxygen isotopes as tracers, Limnol. Oceanogr., 63, S458–S476, doi:10.1002/lno.10775, 2018.